# Serotonin Influences Insulin Secretion in Rat Insulinoma INS-1E Cells

**DOI:** 10.3390/ijms25136828

**Published:** 2024-06-21

**Authors:** Yeong-Min Yoo, Seong Soo Joo

**Affiliations:** 1East Coast Life Sciences Institute, College of Life Science, Gangneung-Wonju National University, Gangneung 25457, Republic of Korea; 2Department of Marine Bioscience, College of Life Science, Gangneung-Wonju National University, Gangneung 25457, Republic of Korea

**Keywords:** serotonin, insulin, serotonin receptor, Akt/ERK

## Abstract

Serotonin or 5-hydroxytryptamine (5-HT) is a monoamine that plays a critical role in insulin secretion, energy metabolism, and mitochondrial biogenesis. However, the action of serotonin in insulin production and secretion by pancreatic β cells has not yet been elucidated. Here, we investigated how exogenous nanomolar serotonin concentrations regulate insulin synthesis and secretion in rat insulinoma INS-1E cells. Nanomolar serotonin concentrations (10 and 50 nM) significantly increased insulin protein expression above the constant levels in untreated control cells and decreased insulin protein levels in the media. The reductions in insulin protein levels in the media may be associated with ubiquitin-mediated protein degradation. The levels of membrane vesicle trafficking-related proteins including Rab5, Rab3A, syntaxin6, clathrin, and EEA1 proteins were significantly decreased by serotonin treatment compared to the untreated control cells, whereas the expressions of Rab27A, GOPC, and p-caveolin-1 proteins were significantly reduced by serotonin treatment. In this condition, serotonin receptors, Gαq-coupled 5-HT2b receptor (Htr2b), and ligand-gated ion channel receptor Htr3a were significantly decreased by serotonin treatment. To confirm the serotonylation of Rab3A and Rab27A during insulin secretion, we investigated the protein levels of Rab3A and Rab27A, in which transglutaminase 2 (TGase2) serotonylated Rab3A but not Rab27A. The increases in ERK phosphorylation levels were consistent with increases in the expression of p-Akt. Also, the expression level of the Bcl-2 protein was significantly increased by 50 and 100 nM serotonin treatment compared to the untreated control cells, whereas the levels of Cu/Zn-SOD and Mn-SOD proteins decreased. These results indicate that nanomolar serotonin treatment regulates the insulin protein level but decreases this level in media through membrane vesicle trafficking-related proteins (Rab5, Rab3A, syntaxin6, clathrin, and EEA1), the Akt/ERK pathway, and Htr2b/Htr3a in INS-1E cells.

## 1. Introduction

Serotonin or 5-hydroxytryptamine (5-HT) is known as a neurotransmitter in the central nervous system and is involved in regulating animal behaviors such as sleep, aggression, and appetite control [1,2]. However, serotonin is present in two distinct parts of the body, the central nervous system and the enteric nervous system located in the gastrointestinal tract. Approximately 90% of serotonin is produced by gastrointestinal enterochromaffin cells [3]. Interestingly, in many species, serotonin is expressed with insulin in pancreatic β-cells and glucagon in pancreatic α-cells [4,5,6,7,8]. The release of extracellular serotonin was associated with stimulating insulin secretion in mouse β-cells [9].

Serotonin is a monoamine and an autocrine signaling molecule that increases pancreatic β-cell function and proliferation [4,10,11,12,13,14]. Serotonin is released with insulin, supporting energy balance and metabolism in pancreatic β cells [8,11,12,15,16,17,18,19,20,21]. During pregnancy or high-fat diet feeding, serotonin maintains glucose homeostasis for energy sufficiency [10,11,22,23].

The effect of serotonin on insulin secretion has been partially elucidated, but the exact mechanism of serotonin action on islets has not yet been established. Currently, two effects of serotonin on target cells are known. The first is the target cell response through the activation of serotonin receptors [4,16], and the second is the target cell response through serotonylation by alternative, endogenous intracellular modification [18]. An increase in serotonin synthesis enhances insulin secretion via the 5-HT2b receptor (or Htr2b) and the 5-HT3 receptor (or Htr3) in pancreatic beta cells [10,11,12,16]. Serotonin also promotes insulin granule exocytosis in a receptor-independent manner through the serotonylation of small GTPase of the Rab family [18]. Serotonin plays a crucial role in regulating body weight, insulin secretion, and glycemic control through the serotonin 2C receptor (or HTR2C) [24,25].

However, the function of serotonin in the intracellular biosynthesis and extracellular secretion of insulin in pancreatic β cells has not been elucidated yet. This study investigated how exogenous nanomolar serotonin concentrations regulate insulin synthesis and secretion in rat insulinoma INS-1E cells with respect to vesicle trafficking-related proteins, Htr2b/Htr3a serotonin receptors, and ERK/Akt phosphorylation.

## 2. Results

### 2.1. Insulin Synthesis and Secretion by Nanomolar Serotonin

We investigated whether serotonin in nanomolar concentrations influenced insulin production and secretion in rat insulinoma INS-1E cells. Serotonin at concentrations of 10 and 50 nM significantly increased insulin synthesis above the levels in the untreated control cells, and 10–100 nM of serotonin decreased the secretion of insulin into the media. At serotonin concentrations of 10–100 nM and after 48 h of treatment, increases and decreases occurred in dose-dependent manners (Figure 1A,B). An autophagy marker, microtubule-associated protein light chain 3 (LC3), did not convert to LC3-I and LC3-II, but protein ubiquitinylation increased according to increases in serotonin concentrations (Figure 1A,C). Therefore, the synthesis of insulin in the cells vs. insulin secretion into the media showed a significant decrease.

### 2.2. Levels of Membrane Vesicle Trafficking-Related Proteins, Serotonin Receptors, and the Serotonylation of Rab3a and Rab27a

The levels of membrane vesicle trafficking-related proteins including Rab5, Rab3a, syntaxin 6, clathrin, and EEA1 were significantly decreased by serotonin treatment compared to the untreated control cells (Figure 2A–G), whereas the expression of Rab27a, GOPC, and p-caveolin-1 proteins were significantly increased by serotonin treatment (Figure 2A,H–J).

Serotonin receptors including the Gαq-coupled 5-HT2b receptor (Htr2b) and the ligand-gated ion channel receptor Htr3a were significantly decreased by serotonin treatment (Figure 3A,C,D), whereas the Gαi protein-associated serotonin receptor 1d (Htr1d) was increased by 10 nM serotonin treatment and decreased by 100 nM serotonin treatment (Figure 3A,B) compared to the untreated control cells.

To confirm the serotonylation of Rab3a and Rab27a by transglutaminase 2 (TGase2) during insulin secretion, we investigated the modified protein levels of Rab3a and Rab27a. TGase2 protein was increased by 10 and 50 nM serotonin treatment in the presence or absence of serotonylation conditions (Figure 4A,B). In serotonylation conditions, Rab3a was serotonylated by treatment with 10 and 50 nM serotonin, but Rab27a was not (Figure 4A,C,D). In particular, Rab3a serotonylation was higher in serotonylation conditions than in the absence of serotonylation conditions (Figure 4A,C).

### 2.3. The Levels of Phospho-ERK, Phospho-Akt, Bcl-2, Bax, and SOD

The levels of the phosphorylation of ERK and Akt were significantly increased by serotonin treatment compared to the untreated control cells (Figure 5). Also, the expression level of the Bcl-2 protein was significantly increased by the 50 and 100 nM serotonin treatments (Figure 6A,B), and the level of the Bax protein was only significantly decreased by the 10 nM serotonin treatment compared to the untreated control cells (Figure 6A,C). The levels of Cu/Zn-SOD and Mn-SOD proteins were significantly decreased by serotonin treatment compared to the untreated control cells (Figure 6A,D,E), whereas the catalase levels were only significantly increased by the 100 nM serotonin treatment (Figure 6A,F).

## 3. Discussion

The relationship between the synthesis and the secretion of insulin may be affected by ubiquitin-mediated protein degradation. Several recent studies demonstrated an essential role of the ubiquitin–proteasome system in β-cells, including insulin synthesis and secretion. Most of the studies used proteasome inhibitors and monitored insulin biosynthesis and secretion. Proteasome inhibition with lactacystin reduced proinsulin synthesis in mouse islets [26]. However, Hartley et al. [27] reported that proteasome inhibition induced endoplasmic reticulum (ER) stress, indirectly reducing insulin biosynthesis. Proteasome inhibition in mouse islets [26] and MIN6 cells [28] inhibited acute glucose-stimulated insulin secretion, whereas lactacystin increased acute glucose-stimulated insulin secretion in rat islets [29]. Although the ubiquitin–proteasome system is essential for maintaining insulin production and secretion, the exact mechanism has not been elucidated.

This study examined the proteins involved in insulin secretion and vesicle trafficking, including Rab5, syntaxin 6, clathrin, EEA1, GOPC, caveolin-1, Rab3a, and Rab27a. In early endosome fusion, GOPC interacts with syntaxin 6, a SNARE and Rab5 effector [30]. Syntaxin 6 also binds to EEA1, another Rab5 effector, suggesting it helps tether post-Golgi vesicles to early endosomes [31]. Cytosolic syntaxin 6 negatively impacts the endosomal system of pancreatic β-cells, indirectly affecting insulin secretion [32]. PI3K forms EEA1-positive early endosomes, which activate Rab5 effectors [33,34,35]. Clathrin is not essential for forming β-cell secretory granules or for proinsulin/insulin exocytosis [36]. Caveolin-1 has explicitly been implicated in insulin secretion and cytokine-induced death [37,38,39]. Caveolin-1 knockout mice were reported to exhibit fasting hyperinsulinemia, insulin resistance, and glucose intolerance [40,41]. Phosphorylated caveolin-1 is associated with the regulation of insulin signaling, including Akt and ERK in pancreatic β-cells [42,43]. Rab3a regulates the replenishment of the readily releasable pool of β-granules by recruiting calmodulin to the resting pool in the early stages of vesicle transport [44]. Rab27a acts directly in the targeting of β-granules from the resting pool to the readily releasable pool at the plasma membrane [45]. In this study, the levels of Rab5, Rab3a, syntaxin 6, clathrin, and EEA1 proteins significantly decreased with serotonin treatment, whereas the expression of Rab27a, GOPC, and p-caveolin-1 proteins was significantly increased by serotonin treatment compared to the untreated control cells. Jung et al. [46] reported that the levels of the Rab5, GOPC, p-caveolin-1, EEA1, and clathrin proteins, excluding the syntaxin-6 proteins, were significantly increased by treatment with nanomolar melatonin concentrations for 48 h compared to control cells without melatonin or glucose treatment.

Serotonin or 5-HT receptors exist in the central and peripheral nervous systems [47], rodent islets [12], and human islets [4,48]. The receptors contain six subfamilies of G protein-coupled receptors (5-HT1-7 or Htr1-7) and one subfamily of a ligand-gated K^+^/Na^+^ ion channel (5-HT3 or Htr3 receptor), which increases or decreases intracellular second messenger cyclic adenosine monophosphate (cAMP) or inositol-1,4,5-trisphosphate (IP3)/diacylglycerol (DAG) to activate an excitatory or inhibitory response. Serotonin increases the number of beta cells through the serotonin receptor 2b (Htr2b) which is associated with the Gαq protein in early pregnancy. It decreases the number of beta cells through the serotonin receptor 1d (Htr1d) associated with the Gαi protein at the end of pregnancy and the postnatal period [4,16]. In contrast, serotonin regulates insulin secretion through Htr3. When serotonin binds to Htr3, Na^+^ ions enter the beta cell, and the membrane potential decreases, facilitating insulin secretion. Under normal metabolic conditions, not all β-cells are needed to maintain normal blood glucose levels. However, metabolic stress such as insulin resistance requires increased insulin secretion. Htr3 can play a critical role in insulin demand in beta cells [49]. No increase in serotonin was observed in Htr3a knockout mice. Also, glucose intolerance appeared in Htr3a knockout mice during pregnancy, and insulin secretion was increased in pancreatic islets by treatment with an Htr3 agonist and decreased by an Htr3 antagonist [11,16]. In this study, Htr2b and Htr3a were significantly reduced by serotonin treatment, whereas Htr1d was increased by 10 nM serotonin treatment and decreased by 100 nM serotonin treatment compared to untreated control cells. Exogenous serotonin treatment of INS-1E cells reduced Htr2b and Htr3a, resulting in a decrease in insulin secretion in response to INS-1E cells.

Serotonylation, the covalent bond between serotonin and proteins, was discovered by Sarkar et al. [50] in the late 1950s, and its mechanism and function have recently been revealed. TGase, such as TGase2 and blood coagulation factor XIIIa, has been shown to catalyze the linkage of serotonin and other monoamines with glutamine residues in the target protein [51,52,53,54,55]. Paulmann et al. [18] demonstrated the importance of serotonylation in β-cells during insulin secretion. The authors showed the serotonylation of Rab3a and Rab27a in RINm5F cells, which is crucial for insulin exocytosis. The authors claimed that extracellular serotonin inhibited insulin secretion by activating Htr1a. A proteasomal inhibitor increased the serotonylated of Rab3a but not Rab27a, showing that proteasomal degradation was associated with the inactivation of Rab3a [56]. The transfection of insulinoma cells with constitutively active Rab3a inhibited insulin release [56], indicating that continuous insulin secretion seems to demand the rapid cycling of Rab3a between an active and inactive state. In contrast, constitutively active Rab27a increased insulin release [57]. In this study, the TGase2 protein was increased by 10 and 50 nM serotonin treatment in the presence or absence of serotonylation conditions. Only Rab3a serotonylation was increased by 10 and 50 nM serotonin treatment in serotonylation conditions.

The PI3K/Akt pathway plays a pivotal role in islet β-cell protection, in which signaling activation enhances islet β-cell survival by stimulating cell proliferation and inhibiting cell apoptosis. The expression of constitutively active Akt linked to an insulin gene promoter in transgenic mice increased the islet β-cells mass by altering the size and number of β-cells. Serotonin exerts part of its action by modulating PI3K/Akt pathway activity [58]. Through G-protein coupled receptors, serotonin activates the PI3K/Akt signaling in pancreatic β-cells [59]. Human Htr1b stimulates the activation of ERK and Akt [60]. Surprisingly, pathways for activating both ERK and Akt were sensitive to two MEK inhibitors at concentrations commonly used to selectively inhibit ERK activation. Both PD098059 and U0126 caused the complete inhibition of ERK phosphorylation and a maximal 60% inhibition of Htr1b-mediated Akt phosphorylation. The inhibition of Akt activation required almost the complete inhibition of ERK.

Insulin signaling in β-cells is crucial for their function. Defective insulin secretion and reduced tissue sensitivity are key features of type-2 diabetes [61,62]. Excessive oxidative stress contributes to mitochondrial and β-cell dysfunction, leading to diabetes. Increased ROS generation by mitochondria is implicated in diabetes pathology [63]. Reducing ROS impact on β-cells could prevent or slow diabetes. Oxidative stress results from excessive ROS production overwhelming the cells’ antioxidant capacity, causing metabolic issues and β-cell damage. Stress-response pathways, including ROS and reduced ATP synthesis, regulate glucose-stimulated insulin secretion [64]. ROS also arise from inflammatory cytokines and bacterial invasion, activating pathways like Akt and ERK [65,66,67,68]. In this study, the level of Bcl-2 protein was significantly increased by treatment with 50 and 100 nM serotonin compared to the untreated control cells, whereas the levels of Cu/Zn-SOD and Mn-SOD proteins decreased.

## 4. Materials and Methods

### 4.1. Cell Culture

The INS-1E cells, which are a clonal pancreatic β-cell line graciously provided by Professor Claes B. Wollheim (University of Geneva, Geneva, Switzerland), were meticulously cultured in RPMI 1640 medium (Invitrogen, located in Carlsbad, CA, USA). This medium was specifically formulated to contain 11 mM glucose and was further supplemented with 10 mM HEPES (pH of 7.3) to maintain optimal physiological conditions. Additionally, the medium was enriched with 10% heat-inactivated fetal bovine serum (FBS) (Invitrogen), ensuring the provision of essential growth factors and nutrients required for cell proliferation. To maintain redox balance within the cells, 50 μM β-mercaptoethanol was included, along with 1 mM sodium pyruvate to support cellular metabolism and energy production. To prevent microbial contamination, the medium was fortified with 50 μg/mL penicillin and 100 μg/mL streptomycin. The cells were cultured at a stable temperature of 37 °C in an incubator set to maintain a humidified atmosphere with 5% CO_2_, thereby simulating physiological conditions conducive to cell growth and function.

For subsequent experimental procedures, the INS-1E cells were cultured in the same RPMI 1640 medium but with a reduced concentration of heat-inactivated FBS, adjusted to 2%. This experimental setup was designed to test the effects of serotonin (Calbiochem in San Diego, MI, USA). The serotonin was administered at varying concentrations of 10 nM, 50 nM, and 100 nM. The cells were then incubated under the same controlled conditions of 37 °C with 5% CO_2_ in a humidified environment for a duration of 48 h. This setup aimed to investigate the potential impact of serotonin on the INS-1E cells over the specified period.

### 4.2. Western Blot Analysis

Cells were meticulously resuspended in a 20 mM Tris-HCl buffer, maintaining a precise pH of 7.4, with this buffer solution being carefully supplemented with 0.1 mM PMSF (phenylmethylsulfonyl fluoride), a serine protease inhibitor. Additionally, the buffer contained 5 µg/mL aprotinin, a broad-spectrum protease inhibitor; 5 µg/mL pepstatin A, an aspartic protease inhibitor; and 1 µg/mL chymostatin, which inhibits chymotrypsin-like serine proteases. Moreover, the buffer was enriched with 5 mM sodium orthovanadate (Na_3_VO_4_) and 5 mM sodium fluoride (NaF), both of which serve as phosphatase inhibitors to prevent dephosphorylation of proteins during the preparation process.

Following the resuspension step, the resulting whole-cell lysate was subjected to centrifugation at a force of 13,000× *g* for a duration of 10 min, with the temperature stringently maintained at 4 °C to preserve protein integrity and prevent degradation. Post-centrifugation, proteins were quantified, and a sample containing 40 µg of proteins or 20 µL of media was carefully prepared for electrophoresis. These samples were then separated using a 10% Gradi-Gel II gradient PAGE (ELPIS-Biotech, Daejeon, Republic of Korea), which allows for the efficient resolution of proteins based on their molecular weight. Once the electrophoretic separation was complete, the proteins were transferred onto a polyvinylidene difluoride (PVDF) membrane, a robust membrane known for its high binding affinity for proteins, facilitating subsequent detection. The PVDF membrane, now containing the transferred proteins, was then incubated with the following antibodies: Insulin (sc-9168), Akt (sc-8321), Bcl-2 (sc-7382), Bax (sc-7380), and β-actin (sc-81178) from Santa Cruz Biotechnology (dilution 1:500) (Santa Cruz, CA, USA); Rab5 (#2143), GOPC (#4646), p-caveolin-1 (#3251), caveolin-1 (#3267), EEA1 (#3288), APPL1 (#3858), syntaxin-6 (#2869), p-mTOR (Ser2448) (#2971), p-mTOR (Ser 2481) (#2974), mTOR (#2983), Raptor (#2280), Rictor (#2114), p-Akt (Thr 308) (#9275), p-Akt (Ser 473) (#9271), p-eIF2α (#3597), eIF2α (#5324), IRE1α (#3294), GRP78/BiP (#3177), and Bcl-xL (#2762) from Cell Signaling Technology (dilution 1:1000) (Beverly, MA, USA). PVDF membrane, now containing the transferred proteins, underwent an incubation process with secondary antibodies. These secondary antibodies were specifically anti-rabbit and anti-mouse IgG conjugated to horseradish peroxidase (Santa Cruz Biotechnology). Following this incubation, the membrane was treated with enhanced chemiluminescence (ECL) Western-blotting reagents (Pierce Biotechnology, Rockford, IL, USA). The visualization process involved exposing the membrane to X-ray film, which captured the chemiluminescent signal emitted by the ECL reagents reacting with the horseradish peroxidase conjugated antibodies. After the appropriate exposure time, the X-ray film was developed, revealing distinct protein bands corresponding to the proteins of interest. To quantitatively analyze the protein bands, the developed X-ray film was scanned using an image scanner, ensuring high-resolution digitization of the bands. The optical density of these scanned protein bands was then measured utilizing the ImageJ analysis software (version 1.37, Wayne Rasband, NIH, Bethesda, MD, USA). Prior to the measurement of optical density, the data were meticulously corrected by performing background subtraction to eliminate any non-specific signal. Additionally, to ensure accuracy and consistency in the protein quantification, the results were normalized by including β-actin as an internal control.

### 4.3. Serotonylation of Rab3a and Rab27a Proteins

INS-1E cells were washed with Hanks’ balanced salt solution, harvested, sonicated at 4 °C for 10 s. And then centrifuged at 10,000× *g* for 5 min. Protein concentrations were determined using the BCA assay (Sigma, St. Louis, MO, USA). Supernatants were analyzed for protein serotonylation. We added 1 mM serotonin to 30 µg of proteins in Tris-containing saline buffer and incubated it at 37 °C for 2 h. SDS-PAGE researched the reaction solution. Serotonylated proteins were analyzed by Western blotting with Rab3A and Rab27A antibodies.

### 4.4. Statistical Analysis

Significant differences were detected by using ANOVA followed by Tukey’s test for multiple comparisons. The analysis was performed using the Prism Graph Pad v4.0 (Graph Pad Software, San Diego, CA, USA). Values are expressed as means ± SD of at least three separate experiments, in which case a representative result is depicted in the figures. *p* values < 0.05 considered statistically significant.

## 5. Conclusions

In conclusion, this study demonstrates that nanomolar concentrations of serotonin (10–100 nM) play a regulatory role in insulin synthesis and secretion in rat insulinoma INS-1E cells. Specifically, serotonin significantly increases insulin protein expression within the cells while decreasing insulin levels in the media, potentially due to ubiquitin-mediated degradation. Serotonin also modulates the expression of membrane vesicle trafficking-related proteins and serotonin receptors, notably reducing Rab5, Rab3A, syntaxin6, clathrin, EEA1, Rab27A, GOPC, and p-caveolin-1 proteins. Furthermore, the serotonylation of Rab3A by TGase2, the increased phosphorylation of ERK and Akt, and the modulation of Bcl-2, Cu/Zn-SOD, and Mn-SOD protein levels collectively suggest a complex regulatory mechanism involving serotonin in insulin metabolism. These findings suggest that the role of serotonin in pancreatic beta-cell function has potential implications for diabetes research. Additionally, the serotonin receptor may be an important target for insulin synthesis and secretion, providing a valuable basis for future studies.

## Figures and Tables

**Figure 1 ijms-25-06828-f001:**
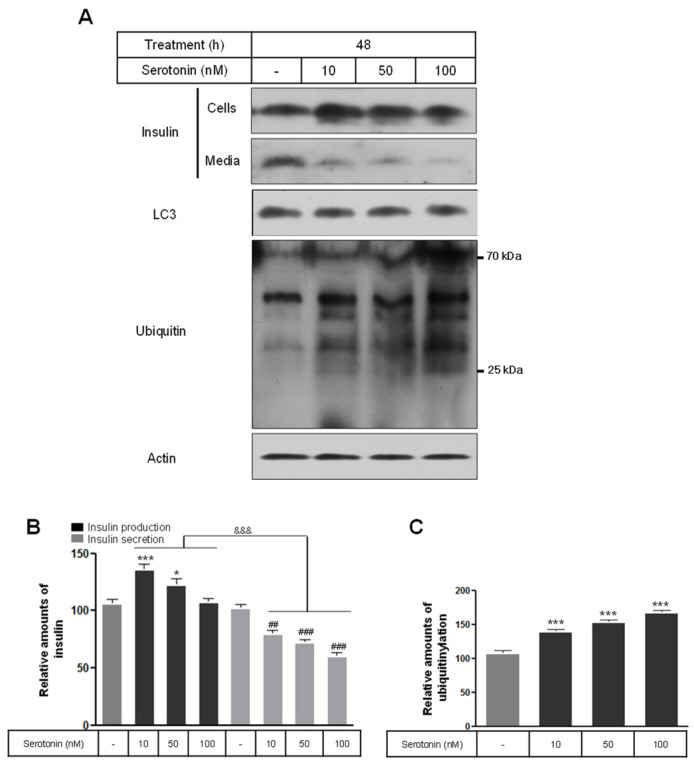
The expression of insulin protein in cells and media in rat insulinoma INS-1E cells treated with serotonin. INS-1E cells were incubated in RPMI 1640 medium supplemented with 2% serotonin for 48 and 72 h at 37 °C with 5% CO_2_. Insulin protein, LC3, and ubiquitin were then analyzed by Western blot (**A**). The relative amounts of insulin protein (**B**) and ubiquitinylation (**C**) were quantified as described in Materials and Methods. Data represent mean ± SD of three experiments. * *p* < 0.05, *** *p* < 0.001 vs. without serotonin in cells; ## *p* < 0.01, ### *p* < 0.001 vs. without serotonin in media; &&& *p* < 0.001, insulin in cells vs. insulin in media.

**Figure 2 ijms-25-06828-f002:**
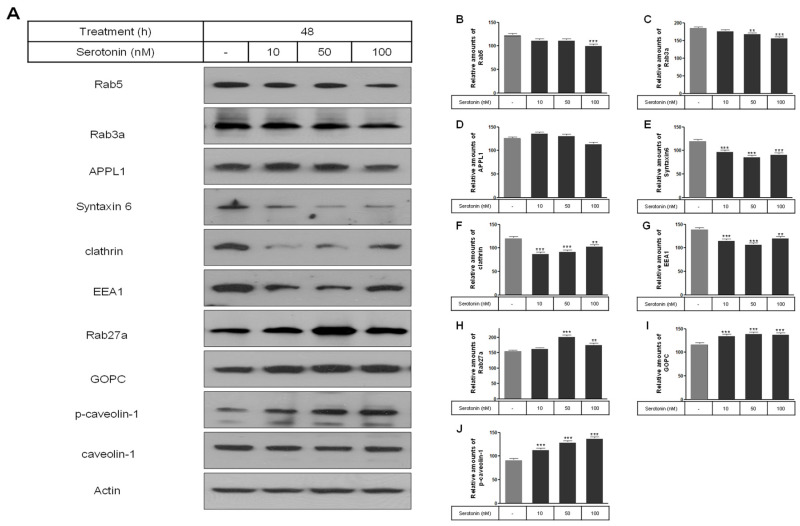
Levels of membrane vesicle trafficking-related proteins including Rab5, Rab3a, APPL1, syntaxin 6, clathrin, EEA1, Rab27a, GOPC, and p-caveolin-1 in rat insulinoma INS-1E cells treated with serotonin. INS-1E cells were incubated in RPMI 1640 medium supplemented with 2% serotonin for 48 h at 37 °C with 5% CO_2_. Proteins were then analyzed by Western blot (**A**). The expressions of membrane vesicle trafficking-related proteins were then analyzed by Western blot (**A**). The relative amounts of Rab5 (**B**), Rab3a (**C**), APPL1 (**D**), syntaxin6 (**E**), clathrin (**F**), EEA1 (**G**), Rab27a (**H**), GOPC (**I**), and p-caveolin-1 (**J**), were quantified as described in Materials and Methods. Data represent mean ± SD of three experiments. ** *p* < 0.01, *** *p* < 0.001 vs. without serotonin in cells.

**Figure 3 ijms-25-06828-f003:**
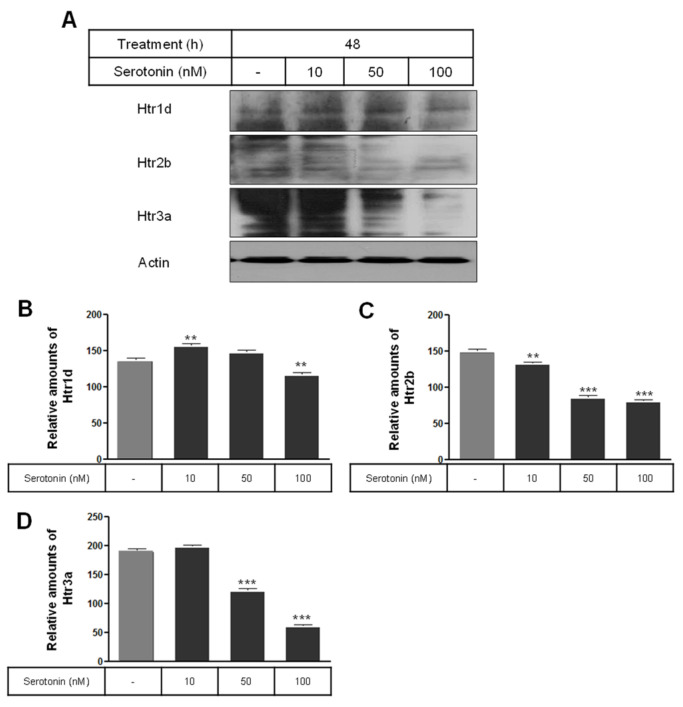
Levels of serotonin receptors Htr1d, Htr2b, and Htr3a in rat insulinoma INS-1E cells treated with serotonin. INS-1E cells were incubated in RPMI 1640 medium supplemented with 2% serotonin for 48 h at 37 °C with 5% CO_2_. The expressions of Htr1d, Htr2b, and Htr3a proteins were then analyzed by Western blot (**A**). The relative amounts of Htr1d (**B**), Htr2b (**C**), and Htr3a (**D**) were quantified as described in Materials and Methods. Data represent mean ± SD of three experiments. ** *p* < 0.01, *** *p* < 0.001 vs. without serotonin.

**Figure 4 ijms-25-06828-f004:**
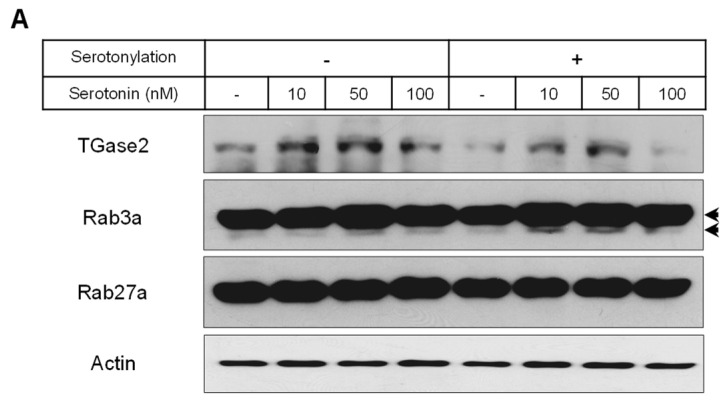
The level of TGase2 protein and the serotonylation of Rab3a and Rab27a in rat insulinoma INS-1E cells. INS-1E cells were incubated in RPMI 1640 medium supplemented with 2% FBS with serotonin for 48 h at 37 °C with 5% CO_2_. TGase2 protein and the serotonylation of Rab3a and Rab27a were then analyzed by Western blot (**A**). The relative amounts of TGase2 protein (**B**) and the serotonylation of Rab3a (**C**) and Rab27a (**D**) were quantified as described in Materials and Methods. Data represent mean ± SD of three experiments. * *p* < 0.05, *** *p* < 0.001 vs. without serotonin in no serotonylation condition; ### *p* < 0.001 vs. without serotonin in serotonylation condition; &&& *p* < 0.001, no serotonylation condition vs. serotonylation condition. Arrows are the serotonylation of Rab3a.

**Figure 5 ijms-25-06828-f005:**
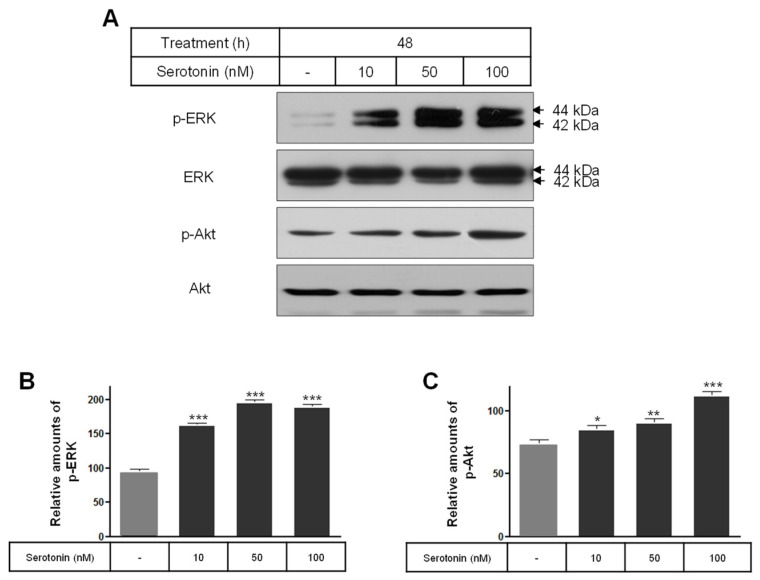
The phosphorylations of Akt and ERK in serotonin-treated rat insulinoma INS-1E cells. INS-1E cells were incubated in RPMI 1640 medium supplemented with 2% FBS with serotonin for 48 h at 37 °C with 5% CO_2_. p-Akt and p-ERK expressions were then analyzed by Western blot (**A**). The relative amounts of p-Akt (**B**) and p-ERK (**C**) were quantified as described in Materials and Methods. Data represent mean ± SD of three experiments. * *p* < 0.05, ** *p* < 0.01, *** *p* < 0.001 vs. without serotonin.

**Figure 6 ijms-25-06828-f006:**
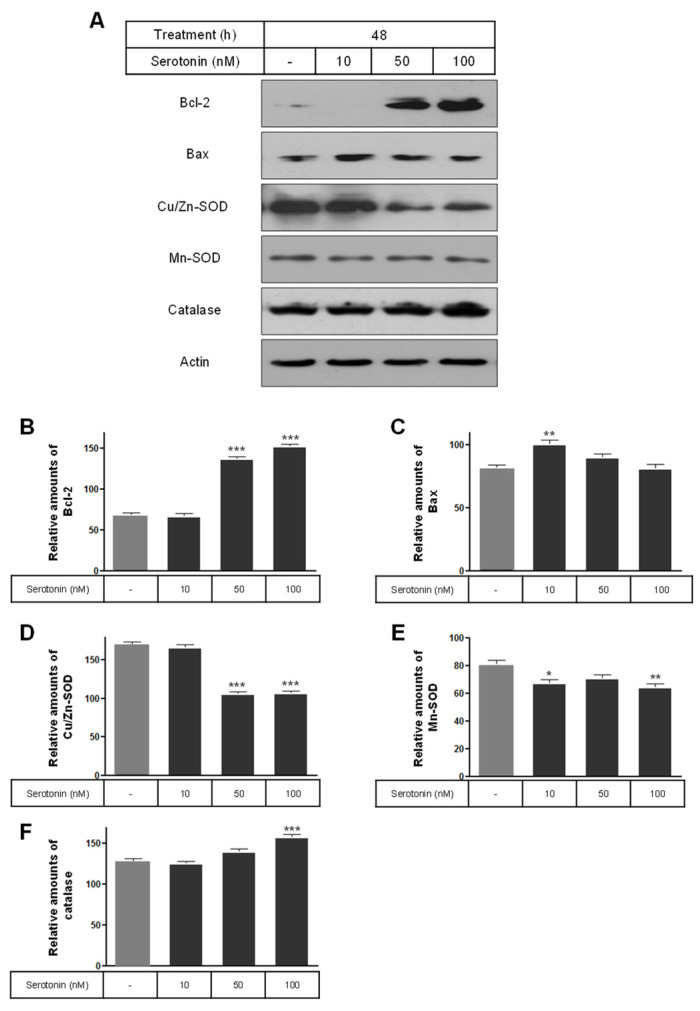
The expression of Bcl-2, Bax, Cu/Zn-SOD, Mn-SOD, and catalase proteins in rat insulinoma INS-1E cells treated with serotonin. INS-1E cells were incubated in RPMI 1640 medium supplemented with 2% FBS with serotonin for 48 h at 37 °C with 5% CO_2_. Bcl-2, Bax, Cu/Zn-SOD, Mn-SOD, and catalase proteins were then analyzed by Western blot (**A**). The relative amounts of Bcl-2 (**B**), Bax (**C**), Cu/Zn-SOD (**D**), Mn-SOD (**E**), and catalase (**F**) proteins were quantified as described in Materials and Methods. Data represent mean ± SD of three experiments. * *p* < 0.05, ** *p* < 0.01, *** *p* < 0.001 vs. without serotonin.

## Data Availability

The original contributions presented in the study are included in the article, further inquiries can be directed to the corresponding authors.

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
