# Peer review of "Serotonin Influences Insulin Secretion in Rat Insulinoma INS-1E Cells"

_ijms, 2024, doi:10.3390/ijms25136828_

Round 1
Reviewer 1 Report
Comments and Suggestions for Authors
Dear Editor, The manuscript is extremely interesting and of great impact in the specific field. However, I believe that it presents some critical issues that need to be addressed. The conclusions are extremely brief, but reflect an overly limited approach to the role of serotonin. We talk about serotonin receptors and therefore about the role of serotonin in influencing the intra- and extracellular secretion of insulin. It would be equally important to verify the role of known agonists and antagonists in modulating this effect. This could also highlight the therapeutic potential that the regulation of insulin secretion through serotoninergic receptors could have. In this way I believe that the work is much more complete and innovative... compared to other work that has already highlighted this role of serotonin Therefore, I believe that the work should be appropriately integrated
Comments on the Quality of English Languagenothing
Author Response
June 7, 2024
Dear Reviewer 1,
Thank you for your letter and for the reviewers’ comments concerning our manuscript entitled " Serotonin influences insulin secretion in rat insulinoma INS-1E cells."
We completely agree with your suggestions regarding our manuscript. The manuscript is completely revised as you and your colleague requested.
The correction and revisions are as follows:
The manuscript is extremely interesting and of great impact in the specific field. However, I believe that it presents some critical issues that need to be addressed. The conclusions are extremely brief, but reflect an overly limited approach to the role of serotonin. We talk about serotonin receptors and therefore about the role of serotonin in influencing the intra- and extracellular secretion of insulin. It would be equally important to verify the role of known agonists and antagonists in modulating this effect. This could also highlight the therapeutic potential that the regulation of insulin secretion through serotoninergic receptors could have. In this way I believe that the work is much more complete and innovative... compared to other work that has already highlighted this role of serotonin Therefore, I believe that the work should be appropriately integrated.
Answer: Thank you very much for your good comments.
We agree with the reviewer that both the serotonin receptor and the role of agonists and antagonists to serotonin are very important studies for serotonin-related insulin secretion. In this study, I wanted to understand the overall mechanisms of serotonin-induced insulin secretion, and in doing so, I presented it in a comprehensive manner by looking at all of them. I hope you understand that, so I have rewritten the conclusion.
The conclusion is restated as follows.
In conclusion, this study demonstrates that nanomolar concentrations of serotonin (10 – 100 nM) play a regulatory role in insulin synthesis and secretion in rat insulinoma INS-1E cells. Specifically, serotonin significantly increases insulin protein expression within the cells while decreasing insulin levels in the media, potentially due to ubiquitin-mediated degradation. Serotonin also modulates the expression of membrane vesicle trafficking-related proteins and serotonin receptors, notably reducing Rab5, Rab3A, syntaxin6, clathrin, EEA1, Rab27A, GOPC, and p-caveolin-1 proteins. Furthermore, the serotonylation of Rab3A by TGase2, the increased phosphorylation of ERK and Akt, and the modulation of Bcl-2, Cu/Zn-SOD, and Mn-SOD protein levels collectively suggest a complex regulatory mechanism involving serotonin in insulin metabolism. These findings emphasize the intricate role of serotonin in pancreatic β cell function and its potential implications for diabetes research.
**All of the edited sections and references were changed with the blue words.
***The Certificate of Editing is attached.
I hope that the revised manuscript is now acceptable for publication in the IJMS. We are looking forward to receiving your answer soon.
Sincerely,
Yeong-Min Yoo Ph.D
College of Life Science,
Gangneung-Wonju National University,
Gangneung, Gangwon-do 25457, Republic of Korea
Email: yyeongm@hanmail.net

Reviewer 2 Report
Comments and Suggestions for Authors
Yoo and colleagues report that serotonin influences insulin levels in INS-1E cells. This was associated with changes in membrane vesicle protein expression and various signalling pathways. Whist the authors presented some clear individual results, they did not connect well together. This lack of a clear narrative makes it had to understand the importance of the work. There are some areas which need inclusion of additional data to be certain of results, and there are other data which I don’t believe are of sufficient quality to be included.
Moreover, these experiments are solely limited to clonal rodent beta-cells lines, so have limited relevance to what might occur in human beta-cells. This manuscript could also do with some improvement to the quality of English writing to help communicate the story.
Please find some more details comments below:
Major issues
· Fig 1 The authors report that serotonin increases insulin protein levels within the cell but also reduced insulin levels in the extracellular media (a marker of secretion). It is on this result that the rest of the paper is based, as such we need to be completely confident in these findings. I would like to see these responses validated by alternative, more quantitative experimental approaches, e.g. qPCR to measure insulin gene expression in response to serotonin, and an insulin secretion assay (a better method of measuring secretion; which would likely be useful to do under glucose stimulated conditions).
· Fig 1 Treatment with serotonin dose dependently increased ubiquitin levels, and the authors inferred this might be responsible for reduced insulin levels seen in the media. I didn’t fully understand this logic, as if insulin was being degraded, I would also expect to see a reduction in intracellular insulin also; however the opposite response was reported. Moreover, there was no direct evidence presented that insulin ubiquitnation was elevated by serotonin treatment, this could have easily been tested using a co-immunoprecipitation approach.
· Fig 3; There are so many bands on the presented blots that I don’t feel they can be interpreted with any confidence. Densitometry analysis certainly cannot be performed on these data accurately. Further, what are the additional blots presented in the original figures and how are these used in figure 3?
· Results throughout; there is no context provided for any of the experiments conducted. A sentence explaining why the authors are moving from one experiment to the next would help the reader understand the story being told.
· Fig 5/6; these data suggested a roll of serotonin in regulating beta-cell viability, as was alluded to in the discussion. Inclusion of a viability assay would have been a helpful way to confirm this.
Minor Issues
· Full blots/original images for figure 1 have not been provided.
· Line 41-42; the authors write ‘…serotonin is expressed with insulin in the islet β-cell, and with glucagon in the islet α-cells…’. This sentence is ambiguous and needs clarifying. Are the authors stating that serotonin is expressed in beta-cells or that it is expressed in hormone granules (or something else)? Either way, more clarity would be helpful.
· All figures; please include a statement in the legends describing the normalisation of densitometry data (instead of as described in materials and methods). My understanding is that this was done using beta-actin for everything, but, for example, was p-caveolin-1 normalised to total caveolin-1?
· Fig 1; no positive control for the induction of autophagy was included in the experiment. This makes interpretation of the results more difficult.
· Fig 2 [B-F]; densitometry analysis of a series of Western blots are presented highlighting changes in expression of various membrane vesicle proteins. These data are representative of three independent experiments. In the experiment depicted, large changes are shown in syntaxin 6 and clathrin expression with serotonin treatment (with a very stable actin expression across treatments). However, densitometry analysis of these data reveal only a small decrease in protein expression, with a tight standard deviation. It is unclear how both these things could be true. Perhaps to be completely transparent all data points should be included on presented bar graphs.
· Line 103; Can the authors please rephrase the first sentence of this paragraph, since it is unclear which ‘condition’ they are referring to.
· Figure 4; I am not familiar with the serotonylation experiment, so my query may be naïve. However, as I understand it, reading through the methods, lysates are treated with serotonin and then prepared for Western blot in the usual way. So, how does probing with Rab3 or Rab27a antibodies guarantee that these proteins would be seratonylated?
· Line 281; Can the authors please include dilutions of the various primary antibodies used for Western analysis?
· Line 307; what does ‘SDS-PAGE researched the reaction conditions’ mean?
Comments on the Quality of English LanguageSee above.
Author Response
June 7, 2024
Dear Reviewer 2,
Thank you for your letter and for the reviewers’ comments concerning our manuscript entitled " Serotonin influences insulin secretion in rat insulinoma INS-1E cells."
We completely agree with your suggestions regarding our manuscript. The manuscript is completely revised as you and your colleague requested.
The correction and revisions are as follows:
Yoo and colleagues report that serotonin influences insulin levels in INS-1E cells. This was associated with changes in membrane vesicle protein expression and various signalling pathways. Whist the authors presented some clear individual results, they did not connect well together. This lack of a clear narrative makes it had to understand the importance of the work. There are some areas which need inclusion of additional data to be certain of results, and there are other data which I don’t believe are of sufficient quality to be included.
Moreover, these experiments are solely limited to clonal rodent beta-cells lines, so have limited relevance to what might occur in human beta-cells. This manuscript could also do with some improvement to the quality of English writing to help communicate the story.
Please find some more details comments below:
Major issues
Fig 1 The authors report that serotonin increases insulin protein levels within the cell but also reduced insulin levels in the extracellular media (a marker of secretion). It is on this result that the rest of the paper is based, as such we need to be completely confident in these findings. I would like to see these responses validated by alternative, more quantitative experimental approaches, e.g. qPCR to measure insulin gene expression in response to serotonin, and an insulin secretion assay (a better method of measuring secretion; which would likely be useful to do under glucose stimulated conditions).
Answer: Thank you very much for your good comments.
This study was conducted in 2020 as part of a comparative paper on the mechanisms of melatonin-induced insulin secretion (Nanomolar melatonin influences insulin synthesis and secretion in rat insulinoma INS-1E cells. J Physiol Pharmacol. 2020;71(5). doi: 10.26402/jpp.2020.5.10.). However, compared to melatonin, serotonin resulted in significantly lower insulin secretion after 2 days of treatment in pancreatic beta cells. Of course, other quantitative measures of insulin secretion as suggested in the reviewer's suggestion would provide a more precise mechanism, but we focused on vesicle trafficking-related proteins, Htr2b/Htr3a serotonin receptors, and ERK/Akt phosphorylation as the reasons for the lower secretion in our results.
It would have been nice to have included in the discussion how the results differed from previous melatonin and serotonin studies, but it was not possible to do so at length.
Previous studies have used both quantitative and fluorescent staining methods to identify melatonin-induced insulin secretion, but we did not present other quantitative methods for serotonin because we only focused on the above-mentioned vesicle trafficking-related proteins, Htr2b/Htr3a serotonin receptors, and ERK/Akt phosphorylation. We apologize for this oversight.
Fig 1 Treatment with serotonin dose dependently increased ubiquitin levels, and the authors inferred this might be responsible for reduced insulin levels seen in the media. I didn’t fully understand this logic, as if insulin was being degraded, I would also expect to see a reduction in intracellular insulin also; however the opposite response was reported. Moreover, there was no direct evidence presented that insulin ubiquitnation was elevated by serotonin treatment, this could have easily been tested using a co-immunoprecipitation approach.
Answer: Thank you very much for your good comments.
We think the reasoning in the review is also reasonable. However, in Figure 1, we are not suggesting that ubiquitin processing is involved in insulin secretion by showing that it can be increased in a concentration-dependent manner by serotonin. Rather, we are suggesting that vesicle trafficking-related proteins, Htr2b/Htr3a serotonin receptors, and ERK/Akt phosphorylation may be involved in the low insulin secretion.
We very much appreciate the methodology you have presented. We will use this method in our comparative study of melatonin and serotonin.
Fig 3; There are so many bands on the presented blots that I don’t feel they can be interpreted with any confidence. Densitometry analysis certainly cannot be performed on these data accurately. Further, what are the additional blots presented in the original figures and how are these used in figure 3?
Answer: Thank you very much for your good comments.
According to Western blot, Htr1d, Htr2b, and Htr3a serotonin receptors were not detected as a single band, but were identified as multiple subtypes corresponding to their size. Therefore, the quantification was plotted as a whole.
Results throughout; there is no context provided for any of the experiments conducted. A sentence explaining why the authors are moving from one experiment to the next would help the reader understand the story being told.
Answer: Thank you very much for your good comments.
We've broken it down as follows
2.1. Insulin synthesis and secretion by nanomolar serotonin
2.2. Levels of membrane vesicle trafficking-related proteins, serotonin receptors, and the serotonylation of Rab3a and Rab27a
2.3. The levels of phospho-ERK, phospho-Akt, Bcl-2, Bax, and SOD
Fig 5/6; these data suggested a roll of serotonin in regulating beta-cell viability, as was alluded to in the discussion. Inclusion of a viability assay would have been a helpful way to confirm this.
Answer: Thank you very much for your good comments.
Viability was checked but not included in the results; all at nanomolar concentrations of serotonin were no different from controls.
Minor Issues
Line 41-42; the authors write ‘…serotonin is expressed with insulin in the islet β-cell, and with glucagon in the islet α-cells…’. This sentence is ambiguous and needs clarifying. Are the authors stating that serotonin is expressed in beta-cells or that it is expressed in hormone granules (or something else)? Either way, more clarity would be helpful.
Answer: Thank you very much for your good comments.
In the Materials and Methods, we stated that ‘Additionally, to ensure accuracy and consistency in the protein quantification, the results were normalized by including β-actin as an internal control’.
We then restated the Materials and Methods
All figures; please include a statement in the legends describing the normalisation of densitometry data (instead of as described in materials and methods). My understanding is that this was done using beta-actin for everything, but, for example, was p-caveolin-1 normalised to total caveolin-1?
Answer: Thank you very much for your good comments.
We presented them in Materials and Methods.
We then restated the Materials and Methods
Fig 1; no positive control for the induction of autophagy was included in the experiment. This makes interpretation of the results more difficult.
Answer: Thank you very much for your good comments.
We suggest that treatment with nanomolar concentrations of serotonin may involve ubiquitination of the entire intracellular protein rather than autophagy.
Therefore, we have removed ‘lines 76-77: suggesting that the media's reduction in insulin protein levels may be associated with ubiquitin-mediated protein degradation’.
Fig 2 [B-F]; densitometry analysis of a series of Western blots are presented highlighting changes in expression of various membrane vesicle proteins. These data are representative of three independent experiments. In the experiment depicted, large changes are shown in syntaxin 6 and clathrin expression with serotonin treatment (with a very stable actin expression across treatments). However, densitometry analysis of these data reveal only a small decrease in protein expression, with a tight standard deviation. It is unclear how both these things could be true. Perhaps to be completely transparent all data points should be included on presented bar graphs.
Answer: Thank you very much for your good comments.
Fig 2 [B-F]; densitometry analysis of a series of Western blots are presented highlighting changes in expression of various membrane vesicle proteins. These data are representative of three independent experiments. In the experiment depicted, large changes are shown in syntaxin 6 and clathrin expression with serotonin treatment (with a very stable actin expression across treatments). However, densitometry analysis of these data reveal only a small decrease in protein expression, with a tight standard deviation. It is unclear how both these things could be true. Perhaps to be completely transparent all data points should be included on presented bar graphs.
Answer: Thank you very much for your good comments.
Rab5, Rab3a, syntaxin 6, clathrin, and EEA1 were decreased and Rab27a, GOPC, and p-caveolin-1 proteins were increased in the overall change by the 10-50 nM concentration treatment. You can see that the overall change is only increased and decreased.
Line 103; Can the authors please rephrase the first sentence of this paragraph, since it is unclear which ‘condition’ they are referring to.
Answer: Thank you very much for your good comments.
We have removed ‘In this condition’, which refers to the 10-100 nM serotonin treatment concentration.
Figure 4; I am not familiar with the serotonylation experiment, so my query may be naïve. However, as I understand it, reading through the methods, lysates are treated with serotonin and then prepared for Western blot in the usual way. So, how does probing with Rab3 or Rab27a antibodies guarantee that these proteins would be seratonylated?
Answer: Thank you very much for your good comments.
Experiments based on reference 18 show that for Rab3 or Rab27a proteins, the changes shown in Figure 4 occur. We looked for other papers, but found very few that were specific to Rab3 or Rab27a. However, our results show that Rab3 undergoes protein changes upon serotonin processing.
Line 281; Can the authors please include dilutions of the various primary antibodies used for Western analysis?
Answer: Thank you very much for your good comments.
We presented them in Materials and Methods.
We then restated the Materials and Methods
Line 307; what does ‘SDS-PAGE researched the reaction conditions’ mean? 'SDS-PAGE
Answer: Thank you very much for your good comments.
We then restated the Materials and Methods
**All of the edited sections and references were changed with the blue words.
***The Certificate of Editing is attached.
I hope that the revised manuscript is now acceptable for publication in the IJMS. We are looking forward to receiving your answer soon.
Sincerely,
Yeong-Min Yoo Ph.D
College of Life Science,
Gangneung-Wonju National University,
Gangneung, Gangwon-do 25457, Republic of Korea
Email: yyeongm@hanmail.net

Reviewer 3 Report
Comments and Suggestions for Authors
The paper is interesting however, the paper needs some improvement before it may be accepted for publication in IJMS:
1. The introduction is too vague - the authors need to clarify the scientific problem and provide some background for the reader who might be not familiar with this field.
2. I think that the authors have a lot higher potential for the analysis of the western blot proteins - i think that based on such comprehensive study, the authors should propose some kind of molecular mechanism (possibly present it on a graph) which would summarize the findings of the paper.
Otherwise, a nice study worth improving.
Author Response
June 7, 2024
Dear Reviewer 3,
Thank you for your letter and for the reviewers’ comments concerning our manuscript entitled " Serotonin influences insulin secretion in rat insulinoma INS-1E cells."
We completely agree with your suggestions regarding our manuscript. The manuscript is completely revised as you and your colleague requested.
The correction and revisions are as follows:
- The introduction is too vague - the authors need to clarify the scientific problem and provide some background for the reader who might be not familiar with this field.
Answer: Thank you very much for your good comments.
Agree with the reviewer's suggestion. I thought the introduction would be too long, so I simply stated it and it became a bit vague, so I rewrote the last part of the introduction.
Lines 61-64: This study investigated how exogenous nanomolar serotonin concentrations regulate insulin synthesis and secretion in rat insulinoma INS-1E cells with respect to vesicle trafficking-related proteins, Htr2b/Htr3a serotonin receptors, and ERK/Akt phosphorylation.
- I think that the authors have a lot higher potential for the analysis of the western blot proteins - i think that based on such comprehensive study, the authors should propose some kind of molecular mechanism (possibly present it on a graph) which would summarize the findings of the paper.
Answer: Thank you very much for your good comments.
This study was conducted in 2020 as part of a comparative paper on the mechanisms of melatonin-induced insulin secretion (Nanomolar melatonin influences insulin synthesis and secretion in rat insulinoma INS-1E cells. J Physiol Pharmacol. 2020;71(5). doi: 10.26402/jpp.2020.5.10.). However, compared to melatonin, serotonin resulted in significantly lower insulin secretion after 2 days of treatment in pancreatic beta cells. So we're looking at the results of the comparative study between melatonin and serotonin to see how they differ, and if there are any mechanisms that might be involved. This research will provide new molecular mechanisms.
**All of the edited sections and references were changed with the blue words.
***The Certificate of Editing is attached.
I hope that the revised manuscript is now acceptable for publication in the IJMS. We are looking forward to receiving your answer soon.
Sincerely,
Yeong-Min Yoo Ph.D
College of Life Science,
Gangneung-Wonju National University,
Gangneung, Gangwon-do 25457, Republic of Korea
Email: yyeongm@hanmail.net

Round 2
Reviewer 1 Report
Comments and Suggestions for Authors
I understand the efforts that have been made by the authors, the conclusions appear improved but I believe that there remains the need to understand at least the effect of serotonin on the receptors and at least on the possible role of agonists and antagonists. Also because it is well known that the interaction of serotonin alone, albeit as an agonist towards the serotonin receptors, could subsequently determine an inhibitory and not a stimulating effect as is actually expected from an agonist Therefore I believe that this aspect still needs to be clarified before the manuscript can be accepted
Comments on the Quality of English LanguageI understand the efforts that have been made by the authors, the conclusions appear improved but I believe that there remains the need to understand at least the effect of serotonin on the receptors and at least on the possible role of agonists and antagonists. Also because it is well known that the interaction of serotonin alone, albeit as an agonist towards the serotonin receptors, could subsequently determine an inhibitory and not a stimulating effect as is actually expected from an agonist Therefore I believe that this aspect still needs to be clarified before the manuscript can be accepted
Author Response
Dear Reviewer 1,
The correction and revisions are as follows:
I understand the efforts that have been made by the authors, the conclusions appear improved but I believe that there remains the need to understand at least the effect of serotonin on the receptors and at least on the possible role of agonists and antagonists. Also because it is well known that the interaction of serotonin alone, albeit as an agonist towards the serotonin receptors, could subsequently determine an inhibitory and not a stimulating effect as is actually expected from an agonist Therefore I believe that this aspect still needs to be clarified before the manuscript can be accepted.
Answer: Thank you very much for your good comments.
Therefore, we have added a section on serotonin receptors and diabetes to the Conclusions section, thanks to your comment.
"These findings suggest that the role of serotonin in pancreatic beta-cell function has potential implications for diabetes research. Additionally, the serotonin receptor may be an important target for insulin synthesis and secretion, providing a valuable basis for future studies."
Please write the following grammatically correct.
*All of the edited sections were changed with the blue words.
I hope that the revised manuscript is now acceptable for publication in the IJMS. We are looking forward to receiving your answer soon.
Sincerely,
Yeong-Min Yoo Ph.D
Reviewer 3 Report
Comments and Suggestions for Authors
The Authors have successfully revised the paper and now I belive it might be accepted for publication.
Author Response
Thank you very much.